# Molecular Characterization of Astrocytoma Progression Towards Secondary Glioblastomas Utilizing Patient-Matched Tumor Pairs

**DOI:** 10.3390/cancers12061696

**Published:** 2020-06-26

**Authors:** Michael Seifert, Gabriele Schackert, Achim Temme, Evelin Schröck, Andreas Deutsch, Barbara Klink

**Affiliations:** 1Institute for Medical Informatics and Biometry (IMB), Carl Gustav Carus Faculty of Medicine, Technische Universität Dresden, D-01307 Dresden, Germany; 2National Center for Tumor Diseases (NCT), Partner Site Dresden, D-01307 Dresden, Germany; gabriele.schackert@uniklinikum-dresden.de (G.S.); achim.temme@uniklinikum-dresden.de (A.T.); Evelin.Schrock@tu-dresden.de (E.S.); barbara.klink@lns.etat.lu (B.K.); 3Department of Neurosurgery, Section Experimental Neurosurgery/Tumor Immunology, University Hospital Carl Gustav Carus, Technische Universität Dresden, D-01307 Dresden, Germany; 4German Cancer Consortium (DKTK), Dresden, German Cancer Research Center (DKFZ), D-69120 Heidelberg, Germany; 5Institute for Clinical Genetics, Carl Gustav Carus Faculty of Medicine, Technische Universität Dresden, D-01307 Dresden, Germany; 6ERN-GENTURIS, Hereditary Cancer Syndrome Center Dresden, D-01307 Dresden, Germany; 7Max Planck Institute of Molecular Cell Biology and Genetics (MPI-CBG), D-01307 Dresden, Germany; 8Center for Information Services and High Performance Computing (ZIH), Technische Universität Dresden, D-01062 Dresden, Germany; andreas.deutsch@tu-dresden.de; 9National Center of Genetics (NCG), Laboratoire national de santé (LNS), L-3555 Dudelange, Luxembourg

**Keywords:** astrocytomas, patient-matched astrocytoma pairs, stage-wise astrocytoma development, secondary glioblastoma, cancer genomics

## Abstract

Astrocytomas are primary human brain tumors including diffuse or anaplastic astrocytomas that develop towards secondary glioblastomas over time. However, only little is known about molecular alterations that drive this progression. We measured multi-omics profiles of patient-matched astrocytoma pairs of initial and recurrent tumors from 22 patients to identify molecular alterations associated with tumor progression. Gene copy number profiles formed three major subcluters, but more than half of the patient-matched astrocytoma pairs differed in their gene copy number profiles like astrocytomas from different patients. Chromosome 10 deletions were not observed for diffuse astrocytomas, but occurred in corresponding recurrent tumors. Gene expression profiles formed three other major subclusters and patient-matched expression profiles were much more heterogeneous than their copy number profiles. Still, recurrent tumors showed a strong tendency to switch to the mesenchymal subtype. The direct progression of diffuse astrocytomas to secondary glioblastomas showed the largest number of transcriptional changes. Astrocytoma progression groups were further distinguished by signaling pathway expression signatures affecting cell division, interaction and differentiation. As expected, IDH1 was most frequently mutated closely followed by TP53, but also MUC4 involved in the regulation of apoptosis and proliferation was frequently mutated. Astrocytoma progression groups differed in their mutation frequencies of these three genes. Overall, patient-matched astrocytomas can differ substantially within and between patients, but still molecular signatures associated with the progression to secondary glioblastomas exist and should be analyzed for their potential clinical relevance in future studies.

## 1. Introduction

Astrocytomas belong to the class of gliomas that represent the most common class of primary malignant human brain tumors in adulthood [1]. Genomic and epigenomic mutations in astrocytes, neural stem cells or neural progenitor cells are most likely contributing to their development, but underlying pathomechanisms are only partly understood [2]. For many years, astrocytomas were classified based on histological features into three groups of increasing malignancy distinguishing between the diffuse astrocytoma WHO grade II (A2), the anaplastic astrocytoma WHO grade III (A3), and the glioblastoma WHO grade IV (G4) [3]. Recently, a combination of histological and genetic features has been established to further improve the classification of astrocytomas [4].

Generally, the vast majority of histologically diagnosed diffuse and anaplastic astrocytomas has a characteristic mutation of the isocitrate dehydrogenase 1/2 gene (IDH1/2), whereas glioblastomas are further distinguished into two genetic subgroups based on the mutational status of IDH [3,5,6,7]. Primary glioblastomas do not have an IDH mutation and are thought to develop within a few months without known precursor states, whereas secondary glioblastomas have an IDH mutation and are known to develop progressively from initial diffuse or anaplastic astrocytomas over several months or years [7]. Secondary glioblastomas are relatively rare compared to primary glioblastomas. About 5% of glioblastomas are estimated to represent secondary glioblastomas based on clinical and imaging data and between 6–13% of glioblastomas are estimated to represent secondary glioblastomas based on the mutational status of IDH [8,9]. It is not possible to distinguish between primary and secondary glioblastomas based on histology, but the absence or presence of an IDH mutation enables this differentiation [4,5]. Such IDH mutations are already present in diffuse or anaplastic astrocytomas that represent precursor lesions of secondary glioblastomas [6,7] and they further induce extensive DNA hypermethylation known as the glioma-CpG island methylator phenotype (G-CIMP) [10,11]. IDH-mutant gliomas also have a better prognosis than IDH wild type tumors [5,12]. Despite advances over the last few years, astrocytomas are not curable with intensive multimodal therapies on the basis of surgery in combination with irradiation and/or chemotherapy.

Large astrocytoma cohorts have been analyzed in comprehensive omics studies over the last years (e.g., [10,13,14,15,16,17,18]), but an in-depth analysis and identification of patterns that drive stage-wise astrocytoma development of individual patients has not been possible by these studies because none or only very few patient-matched pairs of initial and recurrent tumors were available. Nevertheless, these studies enabled the identification of clinically relevant molecular subtypes, frequently mutated genes, and altered cellular pathways. Findings from these and many other studies have also been integrated into a model of temporal molecular gliomagenesis [19].

Initially diagnosed diffuse or anaplastic astrocytomas almost certainly progress to more malignant astrocytomas over several months or years [7], but only little is known about molecular patterns that may drive the stage-wise progression of individual astrocytomas towards secondary glioblastomas. Patient-matched astrocytoma pairs of initial and recurrent tumors offer the great opportunity to study astrocytoma progression based on tumors with a common developmental history overcoming limitations of unmatched astrocytomas of different grades from different patients.

A first milestone in this direction was achieved by the analysis of exome sequencing data of 23 patient-matched astrocytoma pairs [20] suggesting that recurrent tumors originate from very early tumor cells and that temozolomide treatment can influence the evolutionary progression to secondary glioblastomas. Interestingly, 6 of 10 patients that were treated with temozolomide in this study showed an alternative evolutionary path to secondary glioblastomas. The corresponding recurrent gliomas were hypermutated and had potential driver mutations in the RB and Akt-mTOR signaling pathways that gave rise to the signature of temozolomide-induced mutagenesis. In a follow-up study, they also showed that DNA-methylation alterations and somatic gene mutations can both be used independently to reconstruct very similar evolutionary astrocytoma progression trees [21]. They also reported that epigenetic and genomic mutations both converge on the deregulation of the cell cycle. Furthermore, it has been suggested that at least for a subgroup of tumors a reduction of DNA-methylation could influence the stage-wise development of astrocytomas [18]. Still, our knowledge about the stage-wise progression of diffuse or anaplastic astrocytomas to more malignant tumors is far from being complete.

Here, we analyze a unique multi-omics data set of patient-matched astrocytoma pairs of initial and recurrent tumors from 22 patients to gain novel insights into molecular alterations that are associated with the stage-wise development of diffuse or anaplastic astrocytomas to secondary glioblastomas (Figure 1).

## 2. Results

### 2.1. Gene Copy Number Alterations of Patient-Matched Tumor Pairs

We measured gene copy number profiles of patient-matched astrocytoma pairs (A2 to G4: 5, A2 to A3: 1, A3 to A3: 1, A3 to G4: 9, Appendix A) and an unmatched recurrent tumor in comparison to normal reference DNA (Appendix A). Unsupervised hierarchical clustering grouped the tumors into three major subclusters (Figure 2a). The characteristic average gene copy number profiles are shown in Figure 2b,d. We also performed a bootstrap analysis revealing that the three major gene copy number subclusters showed good moderate stabilities and that especially smaller subclusters and patient-matched tumors that directly clustered together were very robust (Appendix A).

Interestingly, only for 7 of 16 patient-matched pairs (43.8%) initial and corresponding recurrent tumors clustered directly together (Figure 2a, green tumor labels). Initial and recurrent tumors of 5 of 16 patient-matched pairs (31.3%) were part of two different major subclusters (Figure 2a). Thus, for more than 50% of patients, the initial and the corresponding recurrent tumor differed from each other like tumors from different patients.

In addition, gene copy number profiles of tumors in the left and right subcluster were relatively heterogeneous in comparison to more homogeneous copy number profiles of tumors of the small middle subcluster (Figure 2a, subclusters: left (orange), middle (yellow), right (red)). Nevertheless, more frequently occurring DNA copy number mutations affecting chromosomal arms or whole chromosomes clearly characterized each subcluster (Figure 2b,d). All A2 were part of the right subcluster, except one recurrent A2 that was part of the left subcluster (P03-A2r). Interestingly, no A2 had a deletion of chromosome 10, but the majority of corresponding patient-matched recurrent tumors showed different types of deletions that affected chromosome 10 (Figure 2a: P04-A2/G4, P05-A2/G4, P06-A2/G4, P18-A2i/G4l). There was also no strong correlation between distances of patient-matched tumors in the gene copy number-based dendrogram and the corresponding times between initial diagnosis and relapse.

Several tumors of the left subcluster showed deletions that affected chromosomal arms 1p, 5q, 6p, 8p, 9p, 11p, 14p, 14q, or 19q or deletions of whole chromosomes 13, 22, or 17 (Figure 2a,b). Some tumors also showed duplications that affected the q-arm of chromosomes 7 and 8 or duplications of chromosome 4 (Figure 2a,b). The majority of tumors in this subcluster were diagnosed as A3 or G4 and only one tumor was diagnosed as A2 (Figure 2a, A2: 1, A3: 8, G4: 6). Initial and recurrent tumors of three patient-matched pairs were directly clustered together (Figure 2a: P03-A2r/G4l, P10-A3/G4, P13-A3/G4). Known cancer genes such as EGFR, EZH2, or CCDN2 were frequently affected by duplications and other known cancer genes such as CDKN2A, RB1, KLK2, or CD79A were frequently affected by deletions. In addition, the majority of tumors had an IDH1 mutation, except four tumors (Figure 2a, P09-A3, P14-A3/G4, P16-A3). Furthermore, the majority were recurrent tumors from patients that were treated with irradiation and/or chemotherapy before surgery (Figure 2a).

All tumors of the small middle subcluster showed a deletion of chromosome 10 in combination with duplications of chromosomes 7 and 19 (Figure 2a,c). Thus, these tumors showed characteristic copy number alterations observed for classical glioblastomas, which is further supported by the observation that none of these tumors had an IDH1 mutation (Figure 2a). In addition, each tumor in this subcluster either showed a deletion that affected the q-arm or the whole chromosome 6 (Figure 2a,c). The four tumors of this subcluster were from two patients whose initial and recurrent tumors directly clustered together (Figure 2a: P07-A3/G4, P12-A3/G4). Additional patient-specific deletions or duplications of chromosomal regions are clearly visible (Figure 2a). Known cancer genes such as EGFR, BRAF, EZH2, and PRDM1 were affected by duplications and CDKN2A, PTEN, RB1, and TP53 were affected by deletions.

Several tumors of the right subcluster showed deletions that affected chromosomal arms 2q, 3p, 4q, 6q, 9p, 10q, or Xp or whole chromosomes 13 and 14 and duplications that affected chromosomal arms 4p, 8q, 10p, or 19p (Figure 2a,d). This subcluster had nearly equal proportions of A2, A3, and G4 (A2: 5, A3: 3, G4: 5) and also contained two patient-matched pairs whose initial and recurrent tumors were directly clustered together (Figure 2a: P22-A2/A3, P11-A3/A4). Known cancer genes such as COL1A1 or STK11 were affected by duplications and other known genes such as CDKN2A, JAK2, PRDM1, FBXW7, or GOLGA5 were affected by deletions in several tumors of this subcluster. In addition, the majority of tumors had an IDH1 mutation, except P09-G4 and P16-G4. Several tumors were treated by irradiation and/or chemotherapy before surgery (Figure 2a).

Globally, there was no co-clustering of gene copy number profiles in relation to prior treatments of tumors before resection (Figure 2a). Tumor samples of patients treated with a combined irradiation and temozolomide chemotherapy were spread over all three major gene copy number subclusters. Furthermore, tumor samples of patients treated with a combined irradiation and non-temozolomide chemotherapy and samples of tumors treated with irradiation were spread over the left and right major subcluster together with tumor samples from untreated tumors. In addition, separate analyses of patients with respect to their treatment revealed that patient-matched gene copy number profiles were directly clustered together for each of the four patients that were treated with a combination of irradiation and chemotherapy after initial tumor resection (Appendix A).

### 2.2. Majority of Patient-Matched Tumors Have Distinct Expression Profiles

We measured gene expression profiles of patient-matched pairs including all pairs that were part of our aCGH analysis and patient-matched tumor pairs of additional patients or multiple relapses (A2 to A2: 2, A2 to A3: 6, A2 to G4: 5, A3 to A3: 1, A3 to G4: 10, Appendix A). Unsupervised hierarchical clustering grouped the individual tumor expression profiles into three major subclusters (Figure 3a). An additional bootstrap analysis revealed that the three major gene expression subclusters had good stabilities and that again smaller subclusters and patient-matched tumors that directly clustered together were very robust (Appendix A).

The composition of the three major gene expression clusters differed strongly from the three major clusters derived based on gene copy number profiles (Figure 3a). Astrocytomas of different WHO grades were spread across all three major gene expression clusters (Figure 3a, left cluster (lilac): 4 A2, 7 A3, 8 G4; middle cluster (light blue): 2 A2, 6 A3, 4 G4; right cluster (green): 7 A2, 4 A3, 3 G4). Only 6 of 25 (24%) patient-matched pairs were directly clustered together (P01, P03, P07, P09, P12, P13). Five of these pairs were part of the gene copy number analysis (Figure 2a) and four of them also showed a direct co-clustering of their gene copy number profiles (P03, P07, P12, P13). This represents a transfer of gene copy number copy alterations to expression levels for the corresponding patient-matched initial and recurrent tumors with very similar gene copy number profiles.

In addition, the majority of IDH1 wild type tumors also tended to form separate subclusters (Figure 3a). This included patient-matched pairs of P07 and P12 with classical glioblastoma-like gene copy number alterations and expression profiles, the direct co-clustering of the P09 pair, and the closely nearby clustering of P14-G4 and P16-G4. Thus, transcriptional alterations distinguishing IDH1 mutant from wild type tumors tend to exist at the genome-wide scale. Again, like for the gene copy number data, there was no significant correlation between the distance of patient-matched tumors in the gene expression-based dendrogram and the time between initial diagnosis and relapse.

Globally, tumor samples of patients treated with a combined irradiation and temozolomide therapy were again spread over all three major gene expression subclusters (Figure 3a). Tumor samples of patients treated with irradiation or with a combination of irradiation and a non-temozolomide chemotherapy did not form larger subclusters. However, the left major gene expression subcluster contained a subcluster of patients that were treated with one of both options, except one sample. Furthermore, additional analyses of patients with respect to their treatment revealed that patient-matched tumor expression profiles were directly clustered together for each of the four patients that were treated with a combination of irradiation and chemotherapy after initial tumor resection (Appendix A). This is in accordance with their clustering of gene copy number profiles (Appendix A).

Next, we determined for each of the three major gene expression clusters under- and overexpressed genes in comparison to normal brain (*q* ≤ 0.01, Appendix A, Figure 3b,c). Compared to the middle cluster, the left and right cluster showed a relatively large number of under- and overexpressed genes that were only exclusively observed for one cluster or commonly altered in both clusters. Generally, most differentially expressed genes encode for transcription factors or co-factors followed by signaling pathway genes (Figure 3b,c). Essential genes, phosphatases, kinases and metabolic pathway genes were significantly overrepresented in some or all major clusters among underexpressed genes. Similarly, overexpressed genes were enriched for tumor suppressors, cancer census genes, kinases, signaling pathway genes, and transcriptional regulators. An additional analysis of individual cancer-relevant signaling pathways showed that the three major clusters mainly differed in the number of genes that were altered in individual pathways (Figure 3d). Most alterations were again observed for the left and right cluster. These two clusters also shared a significant enrichment of differentially expressed MAPK signaling genes. The right cluster showed an exclusive significant enrichment of apoptosis genes and the left cluster an exclusive enrichment of cell cycle genes. No differentially expressed genes involved in DNA replication, base excision, and mismatch repair were observed for the middle cluster.

Finally, we analyzed the expression behavior of cancer census genes in the three major clusters. Genes such as ATM, BCL6, BRCA1, CCNL1, DDX5, MDM4, MAML2, TCF12, and WRN were commonly overexpressed and genes such as BAP1, BCL11A, BCL11B, FOXP1, JAZF1, and SEPT6 were commonly underexpressed in all three major clusters. Exclusively overexpressed genes of the left cluster included FOXO1, JAK1, JAK2, PRDM1, RUNX1, and exclusively underexpressed genes included KIT, MLLT3, PBX1, and SMARCA4. Exclusively overexpressed genes of the right cluster included BUB1B, CCND1, CDK4, EZH2, MYC, NOTCH1, and TP53 and exclusively underexpressed genes included MET and PRKAR1. No exclusively differentially expressed cancer census genes were observed for the middle cluster. In addition, we also specifically analyzed the expression behavior of the MGMT gene. We found that MGMT was underexpressed in all three major gene expression clusters in comparison to the normal brain (*q* < 0.012, Appendix A).

### 2.3. IDH1 Mutant and Wild Type Tumors Differ in Expression of Cancer Genes and Pathways

Motivated by the observation that IDH1 wild type gliomas formed separate subclusters (Figure 3a), we determined genes that differed between IDH1 mutant and wild type astrocytomas. We observed 334 under- and 354 overexpressed genes in IDH1 mutant compared to wild type tumors (Appendix A, *q* < 0.01). This included underexpressed tumor suppressor and cancer census genes such as ALK, CD58, ETV4, FOXO1, FAS, SPRED1, SOCS1, or TNFRSF14 and overexpressed onco- and cancer census genes such as ABL2, ABI1, CSMD3, KCNQ5, MAPK8, MAML2, SUFU, TET1, or RANBP17. The cytokine receptor interaction and the p53 pathway were enriched for underexpressed genes, whereas the Wnt and TGF-Beta signaling pathways were enriched for overexpressed genes in IDH1 mutant tumors. In addition, the pentose phosphate biosynthesis pathway and the valine, leucine, and isoleucine biosynthesis pathway were enriched for underexpressed genes in IDH1 mutant tumors. Furthermore, no significant expression difference was observed for the MGMT gene between IDH1 mutant and wild type astrocytomas (*q* = 0.87, Appendix A).

### 2.4. Initial Tumors Tend to Become Mesenchymal during Progression

We analyzed if the tumors were associated with known glioma subtypes and especially how patient-matched pairs differed. We determined how strongly individual tumors were correlated with the glioma-CpG island methylator phenotype (G-CIMP) [10]. As expected for IDH1 mutant tumors, the vast majority of tumors had moderate to high positive correlations with the G-CIMP subtype (Figure 4a, Appendix A). Three tumor samples with IDH1 wild type showed negative correlations. All other IDH1 wild type tumors showed only weak positive correlations with the G-CIMP subtype.

Next, we performed a classification of tumors as either neural, proneural, classical, or mesenchymal according to the Verhaak classes [14]. The tumors were assigned to three of four classes based on their characteristic correlations profiles that distinguished between proneural, classical, and mesenchymal tumors (Figure 4b, Appendix A). The IDH1 wild type tumors were either assigned to the classical or the mesenchymal class. Furthermore, tumors also formed several separate Verhaak-specific subclusters at the genome-wide expression scale (Figure 3a).

In addition, we analyzed if tumors of patient-matched pairs differed in their assigned Verhaak class (Figure 4c, Appendix A). All patients with initial proneural tumors had recurrent tumors that were classified as mesenchymal. Patients with initial classical tumors had recurrent tumors that were either classical or mesenchymal. The vast majority of patients with initial mesenchymal tumors also had recurrent tumors that were classified as mesenchymal with few exceptions of recurrent tumors that switched to proneural or classical.

### 2.5. Gene Expression Alterations of Astrocytoma Progression Groups

We further determined differentially expressed genes for each patient-matched tumor pair by a specifically designed three-state Hidden Markov Model (HMM) [22]. This was motivated by the presence of deletions and duplications of DNA segments in individual tumor pairs that can lead to similar alterations of gene expression levels of affected genes in close chromosomal proximity. These local chromosomal dependencies were exploited by the HMM to improve the analysis of individual tumor expression profiles [22,23]. We summarized the individual predictions for the different astrocytoma progression groups. In total, we identified 2678 under- and 2087 overexpressed genes for the six patient-matched pairs of the A2 to A3 progression group, 2048 under- and 2160 overexpressed genes for the five pairs of the A2 to G4 progression group, and 4411 under- and 3642 overexpressed genes for the 10 pairs of the A3 to G4 progression group (Figure 5a, Appendix A) that were altered in at least one patient-matched tumor pair of the corresponding progression group.

To determine genes of potential greater relevance for tumor progression, we further restricted this analysis to differentially expressed genes that were altered in at least 50% of the patient-matched tumor pairs of a progression group. This resulted in clearly less differentially expressed genes including 98 under- and 124 overexpressed genes for the A2 to A3 progression group, 78 under- and 219 overexpressed genes for the A2 to G4 progression group, and 136 under- and 59 overexpressed genes for the A3 to G4 progression group (Figure 5b,c, Appendix A). Genome-wide progression group-specific gene expression alteration profiles of these genes are shown in Figure 6. The vast majority of altered genes was exclusively observed in a specific progression group, some genes were altered in the same direction in two progression groups, but none of these frequently altered genes was altered in the same direction in all three progression groups (Figure 5b,c).

A gene annotation analysis revealed that the majority of these frequently altered genes was involved in cellular signaling and transcriptional regulation (Figure 5b,c). Several known cancer genes were among these genes. This included e.g., SDC4 and WIF1 that were under- and BRIP1, BUB1, CASC5, CDKN2C, IL7R, and PDGFRA that were overexpressed in the A2 to A3 progression group, CDKN2A and OMD that were under- and BCL11A, BRIP1, BUB1B, CASC5, CREB3L1, FCGR2B, and WIF1 that were overexpressed in the A2 to G4 progression group, and RET, TSHR, and WIF1 that were under- and COL1A1, FCGR2B, and MET that were overexpressed in the A3 to G4 progression group. In addition, we did not observe MGMT expression changes between individual patient-matched tumors within the astrocytoma progression groups A2 to A3 and A2 to G4 and only observed one down- and one upregulation of MGMT in the A3 to G4 progression group (Appendix A).

Since many of these frequently altered genes were involved in cellular signaling, we performed a more detailed analysis of individual cancer-relevant signaling pathways for the three progression groups (Figure 5d–f). Interestingly, the characteristic signaling pathway gene expression alterations profiles were very similar for the A2 to A3 and the A2 to G4 progression groups (Figure 5d,e). Both groups shared an enrichment of overexpressed cell cycle and ECM receptor interaction pathway genes. Nevertheless, they also differed in the cytokine receptor interaction pathway that was enriched for underexpressed genes in the A2 to A3 progression group and in the p53, PI3K-Akt, and focal adhesion pathways that were enriched for overexpressed genes in the A2 to G4 progression group. In contrast to this, the signaling pathway gene expression alteration profile of the A3 to G4 progression group differed strongly from the two others (Figure 5f). This profile shared the frequent overexpression of p53, PI3K-Akt, focal adhesion, and ECM receptor interaction pathway genes with the other two profiles, but differed in the overrepresentation of overexpressed adherence junction and hedgehog signaling pathway genes.

Finally, we analyzed which of the frequently overexpressed genes could represent potential promising targets for existing drugs or future drug developments utilizing the Open Targets platform [24]. Focusing on genes with the best overall association score, the predicted most promising targets were BRIP1, CDKN2C, and PDGFRA for the A2 to A3 progression group, ATP2B3, BRIP1, NRG1, and VEGFA for the A2 to G4 progression group, and MET for the A3 to G4 progression group. In addition, associations with already existing drugs were reported for PDGFRA, VEGFA, and MET.

### 2.6. IDH1, TP53, and MUC4 Are Frequently Affected by Somatic Single Nucleotide Variations

We performed exome sequencing of patient-matched tumor pairs and corresponding normal blood references (A2 to A3: 3, A2 to G4: 6, A3 to A3: 1, A3 to G4: 6, Appendix A). We determined somatic single nucleotide variations (SNVs, Appendix A) and further analyzed affected genes that were shared between patient-matched glioma pairs or major progression groups (Figure 7). The numbers of shared somatic SNVs between initial and recurrent tumors of patient-matched glioma pairs ranged between 8 and 186 with an average of 60 somatic SNVs shared between both tumors of a patient (Figure 7a). Recurrent tumors showed on average more private somatic SNVs than initial tumors (initial: 26–158, avg. 106; recurrent: 77–396, avg. 173). Considering patient-matched pairs, recurrent tumors showed clearly more private somatic SNVs than their corresponding initial tumors, except for P07-A3/G4 and P15-A3i/A3r (Figure 7a). In addition, the majority of somatic SNVs that affected exons were nonsynonymous substitutions with on average about 29 SNVs per tumor (Figure 7b). Some patient-matched pairs but also especially several recurrent tumors showed increased proportions of stop mutations (Figure 7b). Somatic SNVs were also found in introns with on average about 41 per tumor (Figure 7b).

Focusing on patients that were treated by a combination of irradiation and chemotherapy (Appendix A), we found that the recurrent tumors of P10, P12, and P18 had clearly more somatic SNVs than the corresponding initial tumors, the recurrent tumor of P07 showed similar numbers of somatic SNVs like the initial tumor, and the recurrent tumor of P15 had clearly less somatic SNVs than the initial tumor (Figure 7b). Of those patients with increased numbers of somatic SNVs, only P12 received temozolomide treatment, whereas P10 and P18 received another chemotherapy. In addition, P07 and P15 received temozolomide treatment.

The most frequently mutated gene among our histologically classified tumors was IDH1 closely followed by TP53 (Figure 7c, Appendix A). IDH1 was predicted to be recurrently mutated in 12 of 17 patient-matched pairs. All these pairs had an IDH1 R132H mutation, except IDH1 R132G observed for P03. Except for the initial tumor P08-A3, no IDH1 mutation was predicted in four other patient-matched pairs P01-A2/A3, P07-A3/G4, P12-A3/G4, and P17-A2/A3 of which P07 and P12 were IDH wild type according to initial Sanger sequencing tests. Overall, more than 85% of the initially detected IDH mutations were predicted by exome sequencing.

Three quarters of patient-matched pairs with an IDH1 mutation also had a TP53 mutation (Appendix A). TP53 mutations were also predicted in the patient-matched pairs of P08 and P17. In addition, the initial but not the recurrent tumor of P06 had a TP53 mutation. The most frequent TP53 mutation was TP53 R141C observed in 4 of 10 patient-matched pairs, whereas other nonsynonymous recurrent TP53 mutations affecting exons 1 or 4 were predicted in other pairs.

Moreover, MUC4 was frequently mutated with a mutation frequency similar to that of TP53 (Figure 7c). At least one somatic SNV was predicted in MUC4 in 7 of 17 patient-matched pairs (Appendix A). All predicted SNVs were located in exon 2. The vast majority of these SNVs was nonsynonymous. In contrast to IDH1 or TP53, MUC4 mutations were not always predicted at the same position in patient-matched pairs. Still, for the majority of pairs (6 of 7), the somatic MUC4 SNV was predicted at the same genomic position, in 5 of the 6 pairs, either the initial or the recurrent tumor also showed additional private somatic MUC4 SNVs at other locations with a tendency that recurrent tumors had more additional private MUC4 mutations (recurrent: three additional SNVs in P08-G4 and P05-G4, initial: one additional SNV in P07-A3, P12-A3, and P13-A3). This is supported by somatic MUC4 SNVs predicted in the recurrent tumor samples of P03, P19, and P21, whose initial tumors did not show MUC4 mutations.

Importantly, the variant allele frequencies (VAFs) of the three most frequently mutated genes differed. Median VAFs of IDH1 and TP53 were clearly greater than for MUC4. VAFs ranged from 12.5% to 68.8% with a median of 36.4% for IDH1. TP53 had VAFs in the range of 32.4% to 100% with a median of 69.7%. MUC4 had VAFs in the range of 6.5% to 75% with a median of 15.5%. No SNVs were predicted for IDH1 and TP53 in patient-specific normal blood samples, whereas some SNVs affecting MUC4 with VAFs up to 6.3% were observed in 21% of blood samples.

Specifically focusing on patient-matched pairs, again IDH1, TP53, and MUC4 were the top three of genes affected by somatic SNVs with mutation frequencies that were clearly greater than for all other genes (Figure 7d). We also determined the most frequently mutated genes affected by SNVs considering the three major glioma progression groups (Figure 7e, A2 to A3, A2 to G4, A3 to G4). LNP1 was the most frequently mutated gene exclusively found in patient-matched pairs that progressed from A2 to A3 with a mutation frequency almost twice as large as observed for IDH1 and TP53 in this progression group (Figure 7e). Furthermore, IDH1 was recurrently mutated in each patient-matched pair that progressed from A2 to G4. Characteristics for this progression group were also recurrent TP53 mutations found in more than 60% of tumor pairs and recurrent mutations of MUC4 and TTN found in more than 30% of tumor pairs (Figure 7e). The three most frequently mutated genes in the A3 to G4 progression group were TP53, MUC4, and IDH1. Interestingly, the proportion of MUC4 mutations was as high as that of TP53 and greater than the proportion of IDH1 mutations. Several other genes were also exclusively mutated in more than 30% of patient-matched pairs in this progression group (Figure 7e).

### 2.7. Genes with Small Insertions or Deletions Were Rare, Except for ATRX

We further analyzed the exome sequencing data of the tumors for small somatic insertions and deletions (Appendix A). The top genes most frequently affected by small somatic insertions were DKFZP434A062 and ZBTB46 predicted in 4 of 36 tumors. The top gene most frequently affected by small somatic frameshift deletions was ATRX predicted in 7 of 36 tumors. ATRX also showed stop gain insertions in 3 of 36 tumors. In addition, we determined the most frequent recurrent small somatic insertions and deletions that identically occurred in a specific gene multiple times in different tumors. CCDC57, NCOA3, and UBE2H were each recurrently affected by identical insertions in three different tumors. Similarly, TBX2, MFI2, or MUC12 were each recurrently affected by identical deletions in three different tumors. Importantly, the observed identical MFI2, MUC12, UBE2H, and TBX2 mutations were each observed in a patient-matched tumor pair and an additional tumor sample.

## 3. Discussion

Studies with patient-matched astrocytoma pairs only rarely exist compared to the variety of studies with unmatched astrocytomas from different patients. This has to do with the fact that tumor tissues of sufficient quality have to be collected over many years. Nevertheless, such studies offer the great possibility to better understand astrocytoma progression based on patient-matched tumors with a common developmental history. One of the first studies in this direction were done by Johnson et al. [20], who analyzed exome sequencing data of patient-matched gliomas pairs revealing that recurrent gliomas often developed from very early tumor cells and that temozolomide-induced hypermutations can alter the evolutionary progression. In a follow-up study, Mazor et al. [21] showed that similar evolutionary trees of patient-specific glioma development can be reconstructed independently from DNA methylation alterations and somatic mutations. Still, our current understanding of the progression of individual astrocytomas towards secondary glioblastomas is far away from being complete and most likely also much more complex than we may have thought before. Therefore, we performed a multi-omics data analysis of a unique cohort of patient-matched initial and recurrent astrocytomas to search for mutational and molecular patterns associated with astrocytoma progression. The analyzed tumor pairs were collected from patients that had tumor resections at the Department of Neurosurgery, University Hospital Carl Gustav Carus at the TU Dresden.

Hierarchical clustering of gene copy number profiles revealed three major subgroups, but only 43.8% of initial and their corresponding recurrent tumors clustered together as direct neighbors (Figure 2). Thus, the majority of initial and corresponding patient-matched recurrent astrocytomas differed from each other like astrocytomas from different patients. Factors such as intra-tumor heterogeneity and irradiation and/or chemotherapy after the resection of the initial tumor can contribute to these genomic differences [20,25,26,27,28]. In addition, the purity of tumor samples can have an impact on the clustering. Still, those patient-matched initial and recurrent gliomas that directly clustered together also showed a direct co-clustering at the global gene expression level (Figure 3). Thus, impacts of gene copy number alterations were transferred to the gene expression level. Such a transfer has been known for many years [29] and has also been reported for gliomas [13,22,30].

Globally, there was a trend that gene copy number profiles of A2 were part of the right subgroup, but this subgroup also contained A3 and G4 (Figure 2). Thus, gene copy number profiles of patient-matched pairs could also not be stratified into low and high-grade tumors. Nevertheless, included IDH1 wild type astrocytomas showed characteristic deletions of chromosome 10 and duplications of chromosome 7 that strongly contributed to the formation of a separate major middle subgroup (Figure 2). These tumors can be considered as classical GBMs based on their molecular characteristics [7,17,18].

In addition, more frequent copy number mutations of known caner genes such as deletions of CDKN2A, PTEN, RB1, or TP53 or duplications of EGFR or BRAF also distinguished the left and the middle gene copy number subgroup from the right subgroup (Figure 2). Copy number mutations of CDKN2A and EGFR have been reported as early events in glioma development, whereas PTEN alterations have been described to occur later [25]. In addition, mutations of TP53 are frequently found in gliomas [7,17]. Thus, clinically relevant marker genes differed in their copy number alterations between the subgroups.

Furthermore, none of the low-grade A2 showed a deletion of chromosome 10, but the majority of corresponding recurrent tumors showed deletions on chromosome 10 (Figure 2). This suggests that deletions on chromosome 10 could contribute to the progression of initial low-grade astrocytomas to more malignant recurrent tumors. This is supported by the observation that the loss of heterozygosity of the q-arm of chromosome 10 has already been observed in tumor progression towards secondary glioblastomas in more than 60% of cases [7,31]. One important target of the loss of 10q is the tumor suppressor PTEN [32]. It has also been reported that high-grade glioma patients with 10q deletions benefit from temozolomide treatment [33].

Overall, the three major gene copy number subgroups had good moderate support in terms of stability (Appendix A). The robustness of these major subgroups is limited by the relatively strong heterogeneity of the gene copy number profiles. Since we could only analyze a relatively small data set, the generalization capability also has certain restrictions. Still, the presence of characteristic deletions and duplications in some or most of the tumors of a specific subgroup supports the potential relevance of these major subgroups.

Hierarchical clustering of gene expression profiles also revealed three major subgroups, but they differed strongly from the previously observed gene copy number subgroups, except for the classical GBM-like IDH1 wild type tumors that were part of a smaller subcluster (Figure 3). Generally, astrocytomas of all WHO grades were spread across the three major subgroups. Only 24% of patient-matched initial and corresponding recurrent tumors clustered together as direct neighbors. In addition, the patient-matched astrocytoma expression profiles of the two patients with multiple relapses were quite different. The initial expression profile from P03 differed strongly from both corresponding relapses, and the expression profiles of the four astrocytoma samples from P18 were distributed over all three major gene expression clusters. Thus, patient-matched tumors were more heterogeneous in terms of gene expression changes than in terms of copy number alterations. This suggests that patient-matched astrocytoma expression profiles contain important additional information that are not directly accessible via their gene copy number profiles alone, but which could be important for prognosis and treatment decisions.

The majority of genes affected by expression changes in astrocytomas in comparison to normal brain were transcriptional regulators and signaling pathway genes (Figure 3). Similar to [16], the three major expression subgroups differed mainly in the number of affected genes per signaling pathway with few significant differences between the subgroups affecting the cell cycle and the MAPK and apoptosis pathway. Several known cancer genes such as BRCA1, EZH2, FOXO1, MYC, NOTCH1, or TP53 differed in their expression between the major subgroups. BRAC1 is usually regarded as a tumor suppressor but has also been shown to contribute to tumorigenicity of glioblastoma cells [34]. EZH2 is involved in self-renewal of glioma stem-like cells [35], and its inhibition has been reported as a potential therapeutic strategy for pontine gliomas [36]. FOXO1 is involved in the regulation of cell death and growth of glioma cell lines [37] and has been associated with chemotherapy sensitivity and glial-mesenchymal transition in gliomas [38]. MYC is involved in the regulation of proliferation and survival of glioma cancer stem cells [39]. Inhibition of MCY in glioma cell lines and xenografts has been shown to reduce cell proliferation and to increase apoptosis [40]. NOTCH1 downregulation has been shown to inhibit glioblastoma cell proliferation and neovascularization and to radiosensitize glioblastoma cell cultures and xenografts [41]. Notch signaling is known to be involved in maintenance of glioma stem cells thereby contributing to tumorigenesis and treatment resistance [42]. TP53 is an important clinical marker in gliomas [17] and further known to play complex roles in glioma cells [43]. In addition, MGMT was downregulated in all three major astrocytoma clusters in comparison to normal brain. This is in accordance with prior findings that highlight the potential of MGMT as a prognostic marker for temozolomide treatment [44,45]. Furthermore, MGMT expression was not strongly altered between individual patient-matched tumors and within the vast majority of tumors of the different astrocytoma progression groups further supporting the important role of MGMT as a prognostic marker. Thus, specific expression alterations in individual tumors could potentially provide targets for the development of therapeutic strategies.

Overall, the three major gene expression subgroups were quite robust (Appendix A). The stability of the gene expression clustering is limited by the heterogeneity of the gene expression profiles and the generalization capabilities of the three major clusters are potentially limited because we could only analyze a relatively small data set. Nevertheless, observed expression differences between the three major subgroups and their differences at the level of signaling pathways clearly support their potential relevance.

In addition, IDH1 mutant tumors differed strongly in expression from IDH1 wild type tumors. More than 330 under- and 350 overexpressed genes distinguished both tumor types (Appendix A). These expression differences were also clearly visible at the level of signaling and metabolic pathways. IDH1 mutant astrocytomas showed an overrepresentation of underexpressed genes of the cytokine receptor interaction and the p53 pathway, and an enrichment of overexpressed genes of Wnt and TGF-Beta signaling in comparison to IDH1 wild type tumors. Furthermore, the pentose phosphate and the valine, leucine, and isoleucine biosynthesis pathways were also enriched for underexpressed genes in IDH1 mutant astrocytomas. Such strong differences between both types of astrocytomas can be expected because IDH mutations alter a great variety of molecular processes including energy metabolism, epigenetic reprogramming, cell differentiation, tumor microenvironment, and glioma invasion with impacts on therapy responses [43,46,47].

Furthermore, we observed that IDH1 mutant astrocytomas were positively correlated with the G-CIMP subtype, whereas IDH1 wild type astrocytomas showed either negative or only weak positive correlations (Figure 4). This separation according to the mutational status of IDH1 is expected and further underlines the quality of our analyzed tumor material [10,17]. In addition, we further found that all astrocytomas were either classified as proneural, classical, or mesenchymal, but not as neural according to the Verhaak subtypes [14] (Figure 4). This grouping was also visible in the genome-wide gene expression heatmap by the formation of several subclusters with tumors assigned to the same Verhaak group (Figure 3). Interestingly, with few exceptions, we observed a strong tendency that initial tumors switched to the mesenchymal subtype at recurrence. Such a trend has also been reported by [48]. Mesenchymal gliomas show increased proliferation rates, activation of angiogenesis, inflammation, and higher fractions of necrosis contributing to an increase of malignancy and poor prognosis [14,48]. Thus, the trend of patient-matched tumors to become mesenchymal at recurrence is also in accordance with the increased malignancy of tumors that progress towards secondary glioblastomas.

In addition, we also revealed major expression differences between patient-matched pairs of the three main astrocytoma progression groups. Focusing on genes altered in at least half of the patient-matched pairs of a progression group, we identified clearly more over- than underexpressed genes in the A2 to A3 and A2 to G4 progression groups than in A3 to G4 (Figure 5 and Figure 6). In accordance with greater histologically differences [3,4], the greatest number of differentially expressed genes was observed for the A2 to G4 progression group. The majority of differentially expressed genes was generally involved in transcriptional regulation and cellular signaling. Globally, the A2 to A3 and A2 to G4 progression groups showed very similar patterns of signaling pathway alterations sharing an enrichment of overexpressed genes for cell cycle and ECM receptor interactions. They further showed increased numbers of overexpressed p53, PI3K-Akt, and focal adhesion pathway genes, but they also differed in the cytokine receptor interaction pathway, which was enriched for overexpressed genes in A2 to A3. The A3 to G4 progression group differed markedly from both other progression groups by an enrichment of overexpressed adherence junction and hedgehog signaling genes. Thus, specific signaling pathways were altered in their activity in the individual progression groups and these alterations could influence individual progression paths towards secondary glioblastomas. This could also offer a potential possibility for the development of targeted treatment strategies that take specific signaling pathway alterations into account.

Differential expression of potential key genes potentially contributes to similarities and differences between astrocytoma progression groups. For example, WIF1 was underexpressed in the A2 to A3 and A3 to G4 progression groups, but overexpressed in A2 to G4. WIF1 downregulation has been reported to enhance the migration of glioblastoma cells [49] and to simulate the sensitivity of glioblastomas for Wnt signaling inhibitors [50]. BUB1 was overexpressed in the A2 to A3 and A2 to G4 progression groups. BUB1 inhibition has been reported to decrease proliferation and to improve efficiency of temozolomide and radiation treatment of glioblastoma cells [51]. Other genes such as SDC4 underexpressed in A2 to A3, CDKN2A under- and BCL11A overexpressed in A2 to G4, or COL1A1 and MET overexpressed in A3 to G4 could further contribute to differences between the progression groups. SDC4 has been associated with the outcome of a WT1 immuno therapy in glioblastomas [52]. The deletion of CDKN2A has been associated with poor survival of affected glioma patients [53,54]. BCL11A is mainly known as a transcriptional repressor important for brain development [55]. COL1A1 has been shown to be involved in cell invasion [56], and MET is an important factor involved in a variety of processes that contribute to the malignancy of gliomas [57]. In addition, as indicated by our drug target analysis [24], some of the frequently overexpressed genes in the different astrocytoma progression groups may also offer the potential for targeted treatments with existing drugs or the development of new drugs. This could include targeting of overexpressed PDGFRA, VEGFA, or MET that have already been associated with existing drugs.

We also performed additional heatmap analysis for subsets of patients defined by the specific treatment after initial tumor resection. Despite the fact that only few patients were in each treatment category, we still observed an interesting pattern. For each of the four patients that were treated with a combination of irradiation and chemotherapy, its initial and recurrent tumor were clustered directly together based on their gene copy number and expression profiles. This indicates that a large proportion of molecular alterations present in the initial tumors of these patients were not strongly altered by treatment and therefore again specifically characterized the recurrent tumors. Thus, at least at the global gene copy number and expression scale, the common developmental history of these patient-matched tumors seems to play a greater role than molecular alterations induced by treatment.

Our exome sequencing analysis revealed that patient-matched astrocytoma pairs shared on average 60 SNVs between the initial and the corresponding recurrent tumor (Figure 7). The majority of these SNVs represented nonsynonymous substitutions with on average 29 per tumor. Recurrent tumors showed on average also more private SNVs than initial tumors. Thus, astrocytomas accumulate more mutations that potentially destroy original proteins’ functions during their progression to more malignant tumors. These findings are in good accordance with a study by [20] that also found that initial and corresponding recurrent gliomas differ in their mutational landscape. Different factors such as the origin of recurrent tumors from very early stages of tumor evolution, intra-tumor heterogeneity, irradiation and/or chemotherapy after the resection of the initial tumor, but also purity of tumor samples and quality of sequencing data can contribute to the observed differences between patient-matched pairs [20,25,26,27,28]. As expected, the most frequently mutated genes in our cohort were IDH1 followed by TP53 and 75% of astrocytomas with an IDH1 mutation also had a TP53 mutation. Both genes represent important clinically relevant biomarkers to distinguish between subclasses of gliomas [4,7,17]. The vast majority of IDH1 mutations affected codon 132 with a substitution of arginine to histidine (R132H). The predictions of IDH1 mutations were also in good agreement with initially performed Sanger sequencing, but the observed deviations also indicate limitations of high-throughput analyses. In total, 85% of initially detected IDH1 mutations were also identified by exome sequencing and 100% of astrocytomas without an IDH1 mutation also did not show an IDH1 mutation in exome sequencing. Furthermore, as expected and in good accordance with prior findings [20], we observed that treatment of patients can lead to increased numbers of SNVs in recurrent tumors.

Interestingly, we also found MUC4, which encodes a cell surface associated protein involved in repression of apoptosis and stimulation of proliferation [58], to be affected by SNVs in more than 50% of astrocytomas (Figure 7). The majority of predicted MUC4 SNVs was nonsynonymous and at the same position in both tumors of a patient-matched pair. Locations of these SNVs usually differed between patient-matched pairs, but all SNVs were located within exon 2 of MUC4. We also observed additional MUC4 SNVs in exon 2 private to initial or recurrent tumors with a tendency of more private SNVs in recurrent tumors. Furthermore, we also observed a low background mutation frequency of MUC4 in some of the blood samples of the patient-matched normal blood references. In addition, MUC4 expression levels in astrocytomas were comparable to those of the normal brain references (Appendix A). MUC4 expression levels also did not differ between IDH mutant and wild type tumors. Furthermore, within the different major astrocytoma progression groups, we did not observe MUC4 expression changes for A2 to G4 and A3 to G4, but there was a tendency of reduced MUC4 expression levels for A2 to A3 patient-matched pairs.

Considering the knowledge that has been gained over the years, MUC4 is known to be highly polymorphic between different humans containing numerous sequences of a 48 bp repetitive unit repeated in tandem in exon 2 [59]. However, our usage of patient-matched normal blood references for SNV calling is expected to account for this individual variability. Still, the large gene size of MUC4 could also have contributed to its frequent prediction and we also can not exclude passenger mutations within this gene. Interestingly, different studies have already reported the importance of MUC4 in different cancer types including astrocytomas. For example, MUC4 has been shown to be involved in cell proliferation and invasion by upregulation of EGFR in glioblastoma cells [60]. MUC4 inhibition has been reported to suppress cell growth and metastasis in pancreatic tumor cells [61] and its potential as therapeutic target for pancreatic ductal adenocarcinomas has been reviewed in [62]. MUC4 has been described to promote blood cell association with tumor cells and to be involved in metastasis formation in a breast cancer mouse model [63]. MUC4 expression can further be downregulated by microRNA-150 inhibiting growth, clonogenicity, migration and invasion of pancreatic cancer cells [64]. MUC4 upregulation has been reported to be involved in chemotherapy resistance of gastric cancer cells [65]. Our findings for MUC4 are also supported by the exome sequencing study of patient-matched glioma pairs in [20], where the Appendix A shows that 6 of 23 (26%) patients also had a MUC4 mutation. In contrast to our study, these mutations were almost exclusively observed in recurrent tumors, except for patient 4 of their study that had a deletion that was predicted in the initial and recurrent tumor. In addition, for four of the six patients, all or at least a part of the MUC4 mutations were associated with temozolomide treatment. Thus, chemotherapies could have contributed at least in part to the observation of some of the predicted MUC4 mutations in our cohort, but since the vast majority of our initial tumors was not treated before surgery also chemotherapy-independent mutations of MUC4 could provide an advantage in astrocytoma development. Thus, all these different studies indicate that MUC4 could also play an important role in astrocytoma progression. Still, additional experimental studies are necessary to further evaluate the role of MUC4 in astrocytoma progression and its potential clinical utility.

We also determined the most frequently mutated genes for the three major astrocytoma progression groups (Figure 7). Nothing is known so far for LNP1, which was most frequently affected by SNVs in the A2 to A3 progression group with a mutation frequency nearly twice as large as for IDH1 and TP53. IDH1 was predicted to be mutated in each patient-matched tumor pair of the A2 to G4 progression group. In the A3 to G4 progression group, the mutation frequency of MUC4 was as high as that of TP53 and greater than the mutation frequency found for IDH1. Thus, as known from prior studies [17,18,20], only few genes are affected by recurrent mutations, whereas most mutations only occurred at low frequency or were private for individual tumors.

Finally, we also searched for small insertions or deletions, but they were generally rare among the astrocytomas in our cohort with one major exception. ATRX was frequently affected by small somatic frameshift deletions or stop gain insertions in 28% of the sequenced astrocytomas. ATRX is a clinically relevant marker of lower-grade gliomas [17,66] and has been associated with increased telomere length [18]. ATRX is also involved in epigenetic regulation and its therapeutic potential is currently explored [67].

In summary, patient-matched astrocytomas can differ substantially within and between patients, but still molecular signatures associated with the progression to secondary glioblastomas exist. Future studies should validate these findings in independent cohorts and further analyze their potential clinical relevance.

## 4. Materials and Methods

### 4.1. Patients and Tumor Material

The tumor material was taken from patients that had tumor resections at the Department of Neurosurgery, University Hospital Carl Gustav Carus at the TU Dresden. Patients had given prior informed written consent for the use of their material for research purposes (EK 45032004). The described experimental procedures are in compliance with the ethical standards laid down by the Declaration of Helsinki. An overview of tumor pairs, performed experiments, and treatment of tumors before surgery is provided in Appendix A. Almost all patient-matched tumor pairs can be distinguished into three major categories: (i) no irradiation or chemotherapy after resection of the initial tumor; (ii) irradiation after resection of the initial tumor; and (ii) irradiation in combination with temozolomide or non-temozolomide chemotherapy after resection of the initial tumor (Appendix A).

### 4.2. Gene Copy Number Measurements and Data Analysis

Array-based comparative genomic hybridization (aCGH) was used to compare individual tumor genomes (33 samples including 16 tumor pairs) to normal gender-matched reference DNA (Agilent Euro Male or Female). Experiments were done on Agilent’s SurePrint G3 Human CGH Microarray Kit 2x400K (Design ID: 028081, Agilent) as described previously [68]. Resulting normalized measurements were used to compute an aCGH profile for each tumor. Each aCGH profile comprised 294,370 genomic probes with corresponding log2-ratios quantifying the probe-specific DNA copy number in tumor relative to normal DNA. aCGH profiles were sorted by their chromosomal probe locations and further segmented into chromosomal regions of constant copy number using DNAcopy [69]. Copy number values of 18,808 protein-coding genes (GRCh37.p13) were determined by mapping chromosomal locations of genes to the aCGH segments as described in [70]. Heatmaps were created in R (heatmap.3) to visualize characteristic gene copy number alterations of individual tumors. This was done using one minus Pearson correlation as distance measure in combination with Ward’s clustering algorithm (ward.D2) [71]. A stability analysis of the hierarchical clustering of the gene copy number profiles was performed using the R package pvclust with standard bootstrap settings [72] (Appendix A). Gene copy number values are provided in Appendix A.

### 4.3. Transcriptome Sequencing and Data Preprocessing

Gene expression levels of individual tumors and four commercially available normal brain references were determined by RNA sequencing. Total RNA was extracted from fresh frozen tumor probes with the Qiagen RNeasy Lipid Tissue Mini Kit according to the manufacturer’s protocol. RNA quality was analyzed by an Agilent 2100 Bioanalyzer. Tumor samples with an RNA Integrity Number (RIN) below 7 were excluded from further analysis. mRNA was extracted by Sera-Mag oligo-dT beads and libraries were prepared by the Ultra Directional RNA Lib Prep Kit for Illumina. Sequencing was done on an Illumina HiSeq2000 in two separate runs (first run: 50 bp paired-end for 16 tumor pairs, second run: 100 bp paired-end for 7 tumor pairs, 1 single tumor, and 4 normal brain references). Quality of obtained fastq files was initially checked by FastQC v0.11.4 [73] followed by adapter removal and quality trimming using Trim Galore v0.4.2 [74]. Mapping of reads to the human reference genome (GRCh38 Ensembl release 95) was done using STAR v2.5.3a with standard settings [75] and duplicates were marked using Picard tools v1.141 [76]. Quality analysis of mapped reads was done using RSeQC v3.0.0 [77] to analyze read distributions across gene bodies. Raw read counts per gene were determined by counting gene-specific reads in exons of protein-coding genes using FeatureCounts v1.5.3 [78]. Finally, a gene expression data matrix was created by removing genes without any reads and lowly expressed genes (less than 1 read per million in more than 50% of samples) followed by cyclic loess normalization [79] resulting in normalized log2-counts per million for 14,111 protein-coding genes that were measured in each sample. Gene expression values are provided in Appendix A.

### 4.4. Gene Expression Heatmap and Differential Expression of Major Tumor Clusters

A global gene expression heatmap of all tumor samples was created in R (heatmap.3) to perform an unsupervised analysis of all tumor gene expression profiles. This was done using one minus Pearson correlation as distance measure with distances ranging from zero (two completely identical expression profiles) to one (two completely different expression profiles) as distance measure in combination with Ward’s clustering method (ward.D2) [71]. A stability analysis of the hierarchical clustering of the gene expression profiles was performed using the R package pvclust with standard bootstrap settings [72] (Appendix A). The three major tumor clusters identified by the hierarchical clustering of the tumors in the heatmap were considered to determine differentially expressed genes of each cluster in comparison to the four normal brain reference samples using limma’s standard workflow [79]. All genes with an FDR-adjusted *p*-value (q-value) equal or less than 0.01 were considered as differentially expressed (Appendix A). Corresponding under- and overexpressed genes of the three major tumor clusters were compared among each other by Venn diagrams [80]. Signaling and metabolic pathway annotations and other gene annotations from [70] were used to further analyze the differentially expressed genes and to determine overrepresented categories by Fisher’s exact test. Details to the different annotation sources are provided in Appendix A of [70]. Our cancer signaling pathway analysis was motivated by the KEGG pathways in cancer map (hsa05200). Furthermore, differentially expressed genes and altered pathways between IDH1-mutant and IDH1 wild type tumors were determined using the same computational approach (Appendix A).

### 4.5. G-CIMP and Verhaak Classification

Gene expression log${}_{2}$-ratios of genes in tumor compared to the average expression of four normal brain references were determined for each astrocytoma sample. These log-ratios were used to compute Pearson correlations between the glioma-CpG island methylator phenotype (G-CIMP) gene set (34 of 50 genes were part of the data set) [10] and each of our astrocytoma profiles. Tumors with positive correlations were considered as G-CIMP positive and all other tumors were considered as G-CIMP negative. Similarly, we computed Pearson correlations between the gene expression log-ratios of the Verhaak reference set (735 of 840 genes were part of our data set) [14] and our astrocytoma log-ratio profiles. Each astrocytoma was assigned to the Verhaak class (neural, proneural, classical, or mesenchymal) for which it had the greatest positive correlation. Matched tumor pairs of patient 19 and 20 were excluded from the Verhaak classification because one of their tumors did not correlate well with any Verhaak class. Note that genes missing from the G-CIMP and Verhaak signatures did not strongly influence the correlation analysis because other measured signature genes have strong positive correlations with missing genes and therefore provide a redundant backup [16,81]. G-CIMP and Verhaak analyses results are provided in Appendix A.

### 4.6. HMM-Based Analysis of Patient-Matched Tumor Expression Profiles

A specifically designed three-state Hidden Markov Model (HMM) was used to identify differentially expressed genes in patient-matched tumor expression profiles [22]. The application HMMs for the identification of differentially expressed genes was motivated by the observation that deletions and duplications of DNA segments can lead to positive correlations of gene expression levels of neighboring genes [22,23]. The exploitation of such local chromosomal dependencies has already been shown to improve the prediction of differentially expressed genes for individual breast and brain cancer profiles in comparison to standard log-fold change cutoffs, mixture models, or other more complex models [22,23]. We therefore considered an HMM for our analysis of individual patient-matched astrocytoma expression profiles. We first created a gene expression log${}_{2}$-ratio profile for each patient comparing its recurrent to the corresponding initial tumor based on the normalized tumor expression expression profiles. The resulting data set comprised 25 log-ratio profiles quantifying expression changes of 14,111 genes (Appendix A). Standard settings in combination with initial state-specific means of −3 (underexpressed), 0 (unchanged), and 3 (overexpressed) were used to train a standard first-order HMM on these log-ratio profiles. State-posterior decoding was used to assign each gene in a patient-matched tumor expression profile to its most likely underlying gene expression state (underexpressed, unchanged, or overexpressed) in the recurrent relative to the initial tumor (Appendix A). We summarized under- and overexpressed genes for each astrocytoma progression group and computed Venn diagrams to analyze the intersections between these groups and further analyzed altered genes in the context of signaling pathway annotations from [70].

### 4.7. Exome Sequencing and Data Preprocessing

Somatic gene mutations of individual tumors were determined in comparison to patient-matched normal blood reference samples using exome sequencing. DNA was extracted from fresh frozen probes with phenol-chloroform extraction. Genomic DNA was sheared to approximately 300 bp using the Agilent SureSelect QXT Kit and sequencing libraries were prepared using the Agilent Human All Exon v5 50Mb Kit according to the manufacturer’s recommendations. Sequencing of libraries of tumor pairs and corresponding patient-matched normal blood samples was done in two separate runs on an Illumina HiSeq2000 (first run: 75 bp paired-end for 9 patients; second run: 150 bp paired-end for 9 patients). Quality of obtained fastq files was initially checked by FastQC v0.11.4 [73] followed by adapter removal and quality trimming using Trim Galore v0.4.2 [74]. Mapping of reads to the human reference genome (GRCh37 release 13) was done using BWA-MEM v0.7.13 with standard settings [82] and duplicates were marked using Samblaster v0.1.24 [83]. Local realignment of mapped reads and recalibration of base quality scores was done using the Genome Analysis Toolkit (GATK 3.5, tools: RealignerTargetCreator, IndelRealigner, BaseRecalibrator, PrintReads) [84]. Alignment summary metrics were determined with Picard tools v1.141 [76] and SAMtools v1.3 [85]. Somatic single nucleotide substitutions (SNVs) were determined for each patient-matched tumor normal pair using MuTect v1.1.4 in combination with reference sets of known mutations from dbSNP and Cosmic available from the MuTect developers [86]. Only SNVs that passed all MuTect-filters were further considered and annotated using ANNOVAR v1Feb2016 [87]. We further searched for small somatic insertions and deletions using MuTect2 [88]. Note that we exclude P14 from further analyses because its G4 sample was not enriched for exonic regions. Somatic SNVs and indels predicted for each tumor are provided in Appendix A.

## 5. Conclusions

Patient-matched pairs of initial and recurrent astrocytomas have only rarely been analyzed so far, but they offer the possibility for a more detailed characterization of astrocytoma progression to secondary glioblastomas by taking the common developmental history of patient-specific tumors into account. In this study, we analyzed molecular profiles of patient-matched tumors pairs of 22 astrocytoma patients with and without IDH1 mutations. Interestingly, strongly preserved molecular signatures of such common developmental histories were only found for less than half of the patients in terms of their gene copy number profiles and in less than one-fourth in expression profiles, whereas the other patients had initial and recurrent tumors that differed from each other like unmatched tumors from different patients. Still, important observations like the absence of chromosome 10 deletions in diffuse astrocytomas and their presence in corresponding recurrent tumors, the tendency that recurrent tumors switch to the mesenchymal subtype, or the observation that the common developmental history of initial and recurrent tumors was not strongly altered by treatment for a subset of tumors could be important for prognosis and treatment decisions. Furthermore, expression differences of known cancer genes and signaling pathways distinguishing the three major gene expression subgroups identified in our cohort can add important information that cannot be directly obtained from the gene copy number data. Focusing on astrocytoma progression groups, characteristic expression signatures of signaling pathways regulating cell interaction, differentiation and division, and also differential expression of potential key regulators were predicted to distinguish these groups. In addition, the observation that the three astrocytoma progression groups differ in their frequencies of IDH1, TP53, and MUC4 mutations represents an interesting pattern that may influence the stage-wise progression of diffuse or anaplastic astrocytomas to secondary glioblastomas. Additional studies are necessary to validate our findings and to analyze their potential clinical relevance.

## Figures and Tables

**Figure 1 cancers-12-01696-f001:**
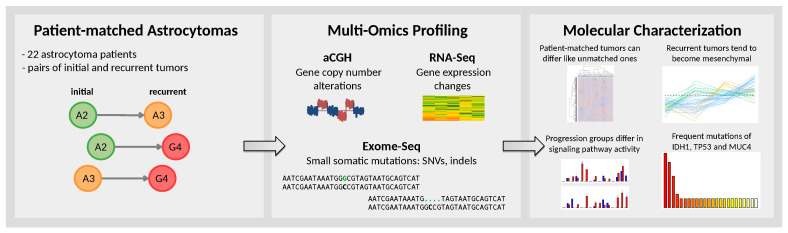
Methodological overview. Left box: A cohort of 22 astrocytoma patients was established to analyze patient-matched pairs of initial and recurrent tumors. Astrocytoma pairs mainly represented patient-specific transitions from diffuse astrocytomas to anaplastic astrocytomas (A2 → A3), from diffuse astrocytomas to secondary glioblastomas (A2 → G4), and from anaplastic astrocytomas to secondary glioblastomas (A3 → G4). Middle box: Comprehensive multi-omics profiling based on array-based comparative genomic hybridization (aCGH), transcriptome sequencing (RNA-Seq), and exome sequencing (Exome-Seq) was performed for the majority of the patient-matched pairs. Right box: Molecular measurements of patient-matched astrocytoma pairs were considered to search for molecular signatures and frequently mutated genes associated with the stage-wise progression of astrocytomas towards secondary glioblastomas.

**Figure 2 cancers-12-01696-f002:**
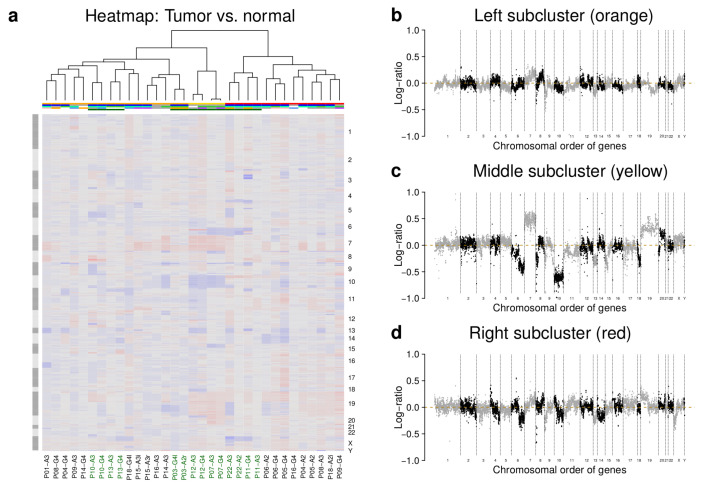
Genome-wide gene copy number alterations of individual astrocytomas. (**a**) heatmap of individual tumor aCGH profiles. Tumors were clustered based on the similarity of their aCGH profiles (columns) and their corresponding gene-specific copy number values are visualized from chromosome 1 to X (rows). The heatmap represents gene deletions (blue regions), gene duplications (red regions), and genes with unchanged copy number (grey regions) in tumors compared to normal DNA. Three major tumor-specific aCGH clusters are labeled in the first colored line below the dendrogram at the top of the heatmap (left subgroup: orange, middle subgroup: yellow, right subgroup: red), the second line represents the IDH1 mutation status (mutated: blue, wt: grey), the third line represents the estimated tumor content of the sample (grey: less than 25%, green: 25–75%, light green: more than 75%, white: not available), the fourth line contains information about treatment before surgery (grey: no, pink: surgery, orange: surgery and irradiation, lilac: surgery, irradiation and temozolomide, blue: surgery, irradiation and non-temozolomide chemotherapy, white: not available), and patient-specific initial and recurrent tumors that directly clustered together are labeled in green in the fifth line. Alternating dark and light grey bars left to the heatmap visualize the individual chromosomes. Tumor identifiers are shown below the heatmap encoding the patient identifier (Px: x specifies the patient) followed by the tumor type (A2: astrocytoma WHO grade II, A3: astrocytoma WHO grade III, G4: glioblastoma WHO grade IV; additional extensions of ‘i’ (initial), ‘r’ (recurrent), and ‘l’ (last recurrent) are added when a patient had more than two tumors). (**b**–**d**) characteristic average aCGH profiles of the three major aCGH subgroups. Gene copy number alterations are quantified by average log2-ratios of tumor versus normal and plotted in chromosomal order of genes. Chromosomes are separated by alternating grey and black dots and chromosome ends are marked by black dotted vertical lines. Strong deviations of average log-ratios from zero indicate deletions (negative log-ratio) and duplications (positive log-ratio) in the left subgroup (**b**), middle subgroup (**c**), and right subgroup (**d**).

**Figure 3 cancers-12-01696-f003:**
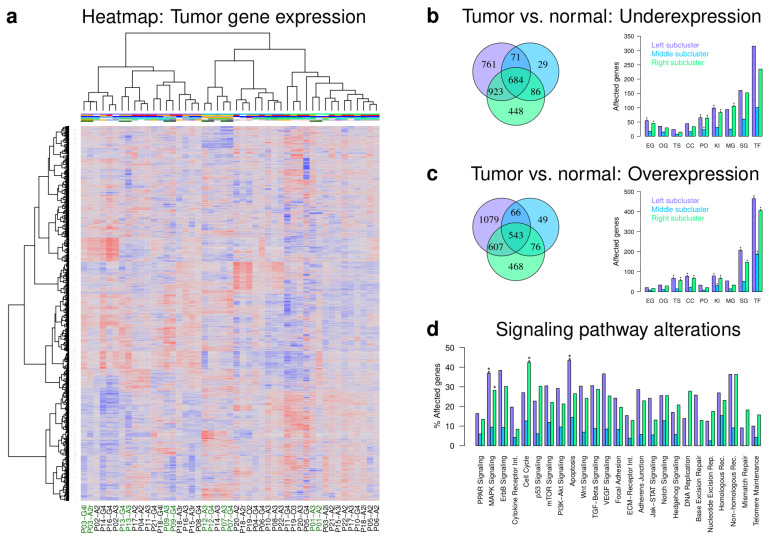
Genome-wide gene expression profiles of individual astrocytomas. (**a**) heatmap of individual tumor gene expression profiles. Tumors were clustered based on the similarity of their gene expression profiles (columns) and their corresponding gene-specific expression values are visualized (rows). The heatmap represents z-score scaled expression levels of each gene with values clearly less than zero in blue shades, values about zero in grey, and values clearly greater than zero in red shades. Three major tumor-specific gene expression clusters are labeled in the first colored line below the dendrogram at the top of the heatmap (left subgroup: lilac, middle subgroup: light blue, right subgroup: green), previous assignments of tumors to the three major aCGH clusters in Figure 2 are shown in the second line, the third line represents the IDH1 mutation status (mutated: blue, wt: grey), the fourth line represents the classification of the tumors according to the Verhaak classes (blue: mesenchymal, orange: classical, green: proneural, white: unclassifiable), the fifth line represents the estimated tumor content of the sample (grey: less than 25%, green: 25–75%, light green: more than 75%, white: not available), the sixth line contains information about treatment before surgery (grey: no, pink: surgery, orange: surgery and irradiation, lilac: surgery, irradiation and temozolomide, blue: surgery, irradiation and non-temozolomide chemotherapy, white: not available), and patient-specific initial and recurrent tumors that directly clustered together are labeled in green in the seventh line. Light and dark grey lines right to the left dendrogram of the heatmap highlight known signaling pathway and cancer genes. Tumor identifiers are shown below the heatmap encoding the patient identifier (Px: x specifies the patient) followed by the tumor type (A2: astrocytoma WHO grade II, A3: astrocytoma WHO grade III, G4: glioblastoma WHO grade IV, O2: oligoastrocytoma WHO grade II, O3: oligoastrocytoma WHO grade III; additional extensions of ‘i’ (initial), ‘r’ (recurrent), and ‘l’ (last recurrent) are added when a patient had more than two tumors); (**b**,**c**) comparison of differentially expressed genes distinguishing the three major gene expression clusters. Tumors of each major cluster were compared to four normal brain reference samples. The Venn diagram (left) shows similarities and differences between differentially expressed genes at the FDR-level of 1% and the barplot (right) shows corresponding numbers of genes that are part of specific cancer-relevant gene annotation groups (EG: essential gene, OG: oncogene, TS: tumor suppressor gene, CC: cancer census gene, PO: phosphatase, KI: kinase, MG: metabolic gene, SG: signaling gene, TF: transcription factor or co-factor). Enrichment of genes in an annotation category is shown by asterisks (Fisher’s exact test: ‘**’: *p* < 0.01, ‘*’: *p* < 0.05). Underexpressed genes are shown in (**b**) and overexpressed genes are shown (**c**); (**d**) proportions of genes of specific signaling pathways affected by differential expression within the three major gene expression clusters. Under- and overexpressed genes at the FDR-level of 1% were considered and overrepresented pathways are marked by asterisks (Fisher’s exact test: ‘**’: *p* < 0.01, ‘*’: *p* < 0.05).

**Figure 4 cancers-12-01696-f004:**
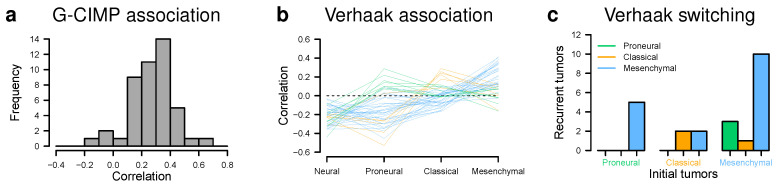
Associations of tumor expression profiles with G-CIMP and Verhaak subtypes. (**a**) Pearson correlations of tumor gene expression profiles with the corresponding genes in the G-CIMP reference set. Tumors with positive correlations are G-CIMP positive and tumors with negative correlations are G-CIMP negative; (**b**) Pearson correlations of tumor gene expression profiles with the corresponding genes in the Verhaak classification reference set. Each tumor is represented by a curve that shows tumor-specific associations with the four Verhaak subtypes (neural, proneural, classical, mesenchymal). Each tumor was assigned to the Verhaak class with the greatest positive correlation and the corresponding tumor-specific correlation curve is colored to visualize this assignment (green: proneural, orange: classical, blue: mesenchymal); (**c**) Verhaak classes assigned to patient-matched tumor pairs. Initial tumors were observed to fall into three Verhaak classes (*x*-axis: proneural, classical, mesenchymal) and corresponding assignments of recurrent tumors to one of the three Verhaak classes are quantified by the specific colored bars.

**Figure 5 cancers-12-01696-f005:**
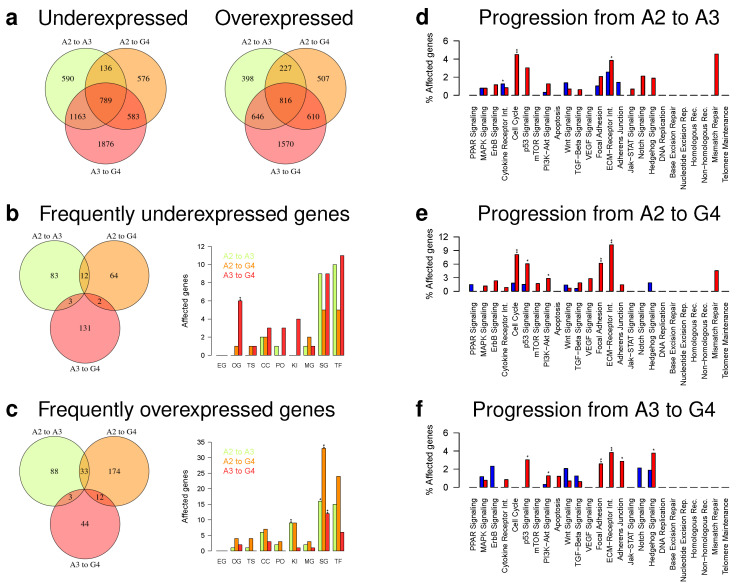
Differentially expressed genes and signaling pathway alterations distinguishing astrocytoma progression groups. Differentially expressed signaling and metabolic pathway genes were determined for each patient-matched astrocytoma pair in each of the three progression groups (A2 to A3, A2 to G4, A3 to G4). (**a**) Venn diagrams of under- and overexpressed genes observed in at least one patient-matched tumor pair; (**b**,**c**) Venn diagrams and bar plots of functional annotations of under—(**b**) and overexpressed (**c**) genes observed in at least 50% of patient-matched pairs of a progression group. Bar plots represent cancer-relevant gene annotation groups (EG: essential gene, OG: oncogene, TS: tumor suppressor gene, CC: cancer census gene, PO: phosphatase, KI: kinase, MG: metabolic gene, SG: signaling gene, TF: transcription factor or co-factor). Enrichment of genes in an annotation category is shown by asterisks (Fisher’s exact test: ‘**’: *p* < 0.01, ‘*’: *p* < 0.05); (**d**–**f**) Average proportions of underexpressed (blue) and overexpressed (red) individual signaling pathway genes observed in at least 50% of patient-matched pairs of a progression group. Enrichment of genes for an individual pathway is shown by asterisks (Fisher’s exact test: ‘**’: *p* < 0.01, ‘*’: *p* < 0.05).

**Figure 6 cancers-12-01696-f006:**
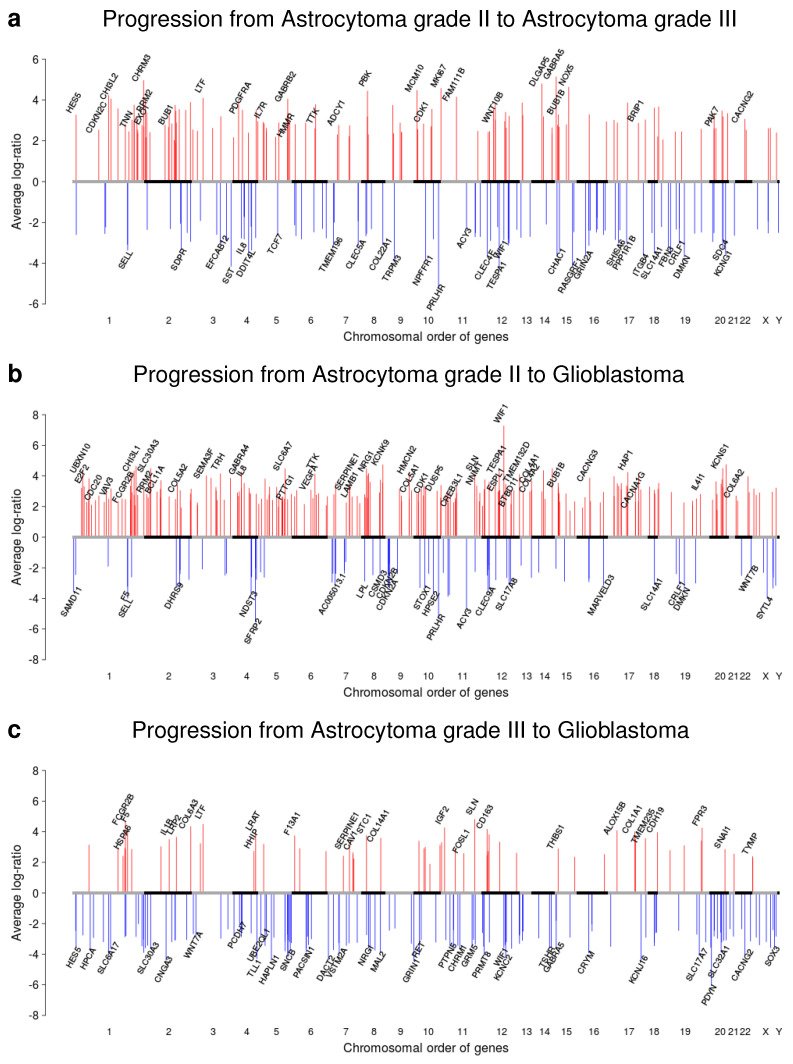
Genome-wide average expression profiles of frequently altered genes in astrocytoma progression groups. Average gene expression profiles of under- (blue) and overexpressed genes (red) observed in at least 50% of patient-matched astrocytoma pairs of a progression group (A2 to A3 (**a**); A2 to G4 (**b**); A3 to G4 (**c**)). The expression alteration of each under- or overexpressed gene (*y*-axis) is quantified by the average log${}_{2}$-ratio of recurrent to initial tumor expression measurements considering each patient-matched tumor pair that had the corresponding gene expression alteration in a specific progression group. Average expression log-ratios of genes are shown in chromosomal order along the *x*-axis from chromosome 1 to Y. Gene names of some strongly altered genes are shown with a specific focus of genes involved in signaling and metabolic pathways and known cancer genes.

**Figure 7 cancers-12-01696-f007:**
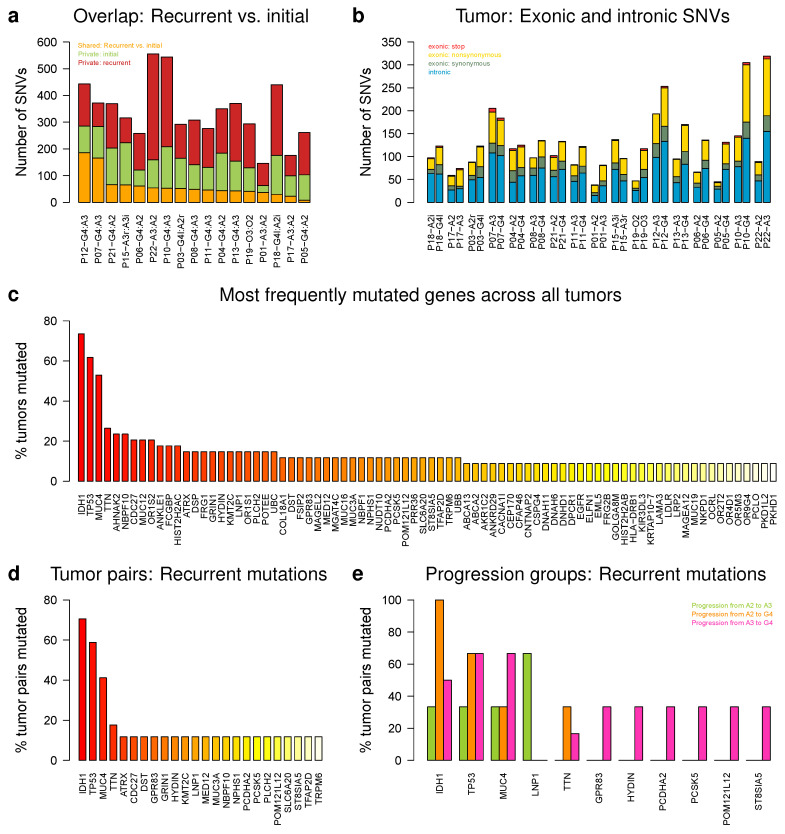
Overview of predicted somatic single nucleotide variations and frequently mutated genes. Somatic SNVs were determined for each patient-matched tumor-normal pair by exome sequencing. Somatic SNVs were further analyzed for individual astrocytomas, patient-matched astrocytoma pairs, and astrocytoma progression groups. (**a**) numbers of somatic SNVs shared between the initial and the recurrent tumor of a patient-specific astrocytoma pair (orange) and corresponding numbers of somatic SNVs that were private to the initial tumor (green) or the recurrent tumor (red); (**b**) numbers of exonic and intronic SNVs predicted for the initial and the recurrent tumor of each patient-specific astrocytoma pair; (**c**) most frequently mutated genes affected by somatic SNVs across all astrocytoma samples; (**d**) most frequently mutated genes shared between the initial and the recurrent tumor of patient-matched astrocytoma pairs; (**e**) most frequently mutated genes shared between the initial and the recurrent tumor of patient-matched astrocytoma pairs of a progression group.

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
