# Peer review of "Molecular Characterization of Astrocytoma Progression Towards Secondary Glioblastomas Utilizing Patient-Matched Tumor Pairs"

_cancers, 2020, doi:10.3390/cancers12061696_

Round 1

Reviewer 1 Report

General: The authors compare patient-matched astrozytoma pairs of initial and recurrent tumors from 22 patients. They characterize difference in terms of CNVs, differential gene expression and Verhaak subtypes. 

Major:

  1. A general limitation of the study is that it only comprises 22 patients. This raises concerns regarding the statistical stability of the simple hierarchical cluster analysis for gene expression data and CNVs. The choice of 4 and 3 clusters, respectively, seems rather arbitrary. I recommend a resampling based clustering approach, using either a bootstrap (as e.g. implemented in R-package pvclust) or Monti's consensus clustering method (e.g. R-package ConsensusClusterPlus).
  2. Which pathway database was used for enrichment analysis? Was a hypergeometric test used? Please clarify.
  3. What's the rationale behind the HMM based comparison of patient-matched gene expression profiles? From my perspective there are no hidden states, because over-, under- and unchanged expression are directly observable in gene expression. So, we are left with a multi-state model. Moreover: How do the authors consider the time passed between initial diagnosis and recurrence in their model?
  4. Which potential therapeutic implications do the authors see as a consequence of their analysis? For example, do some of the pathways altered during the progression of the disease contain currently discussed drug targets in Open Targets?

Author Response

Please consider our attached PDF file for the responses to your comments. They are given under Reviewer 1.

Reviewer 2 Report

Seifert et al performed a molecular characterization of astrocytoma progression towards secondary glioblastomas but some points need to be improved:

1) some cases have low grade astrocytoma (grade 2 astrocytoma) as recurrent tumors and not anaplastic astrocytoma or glioblastoma and so these cases should be deleted from the global genetic analysis

2)the treatment received by patients can modify genetic alteration; the cases described in this paper reported different treatment, i.e. some patient were treated with radiotherapy others with chemotherapy alone. The authors should separate at least patients receiving chemotherapy from other patients; moreover, the type of chemotherapy should be explained because temozolomide was demonstrated to have a strong power to alter genes.

3) Could you perform a survival analyses based on different genetic alteration? you should show if there is a specific gene alteration impacting on survival

4) Shortens the abstract and use specific paragraphs such as introduction, methods, etc...

Author Response

Please see the attached PDF file for our responses to your comments. They are listed under Reviewer #2.

Reviewer 3 Report

The study entitled “Molecular characterization of astrocytoma progression towards secondary glioblastomas utilizing patient-matched tumor pairs” is a novel and interesting work with an exhaustive molecular analysis about the principal elements implicated in the progression of astrocytomas grade II and III toward glioblastomas. The analysis of data from patient-matched tumor pairs together with finding the specific genes implicated in the progression of astrocytomas are invaluable results that could have a great impact on the future development of more effective therapies against these tumors. However, there are some aspects that authors could consider to improve the quality of the article.

Comments:

Abstract:

Define the three astrocytoma expression groups.

Keywords:

The authors should consider including the word “secondary glioblastoma” since it is an essential element in all work.

Introduction: Give information about the incidence of primary and secondary glioblastomas.

Mention the role of IDH1 mutations in glioblatomas progression.

Clarify the influence of temozolomide in the progression to secondary glioblastomas.

The authors say: “Genomic and epigenomic mutations in astrocytes or neural stem cells are most likely…” the authors should consider including in this sentence the term neural progenitor cell since they have also been associated with the development of glioblastoma.

Results:

Explain the transition A3 to A3.

Lines 104 and 105: the authors say “... majority of tumors in this subcluster were diagnosed as A2 or G4 and only one tumor was diagnosed as A2” the author should say “as A3 or G4”.

Increase the intensity of the colors in the heatmaps of Figures 2a and 3a. They cannot be distinguished.

Could the authors explain why is missing the pair of patient P01-A3 in the heatmap of Figure 2a?

In the title of Figures 6b and c the authors should consider changing the term glioblastoma grade IV to astrocytoma grade IV or just glioblastoma since the latter implies by itself that it is grade IV.

Figure Legend 6: the correct in (b) is: A2 to G4.

Figure 7e: The correct is: A2 to G4.

Discussion:

Lines 370 and 371: the authors say “Thus, patient-matched tumors were more heterogeneous in terms of gene expression changes than in terms of copy number alterations”, what implications could this have?

In this study the authors did not find significant variation in the MGMT expression, considering that the promoter of this gene is hypermethylated in 75 % of secondary glioblastomas, the authors should discuss this fact.

Conclusions:

Taking into account the high volume of results obtained in this study the authors should consider include more elements in the conclusions.

Author Response

Please see the attached PDF file for our responses to your comments. They are listed under Reviewer #3.

Round 2

Reviewer 1 Report

The authors adequately addressed all my concerns.

Reviewer 2 Report

the authors improved the text according to the suggested recommendations.